**Data Availability Statement:** All data is available via OSF: https://osf.io/qsua8/files/?view_only=ea7a550400f4403eb280b1d383ec11fa.

**Funding:** The author(s) received no specific funding for this work.

# The effect of librarian involvement on the quality of systematic reviews in dental medicine

**Jana Schellinger**[1], **Kerry Sewell**[2], **Jamie E. Bloss**[2]*, **Tristan Ebron**[3], **Carrie Forbes**[2]

**1** Emory & Henry College School of Health Sciences, Marion, VA, United States of America, **2** Laupus Health Sciences Library, East Carolina University, Greenville, NC, United States of America, **3** North Carolina Department of Natural and Cultural Resources, Raleigh, NC, United States of America

* blossj19@ecu.edu

## Abstract

### Objectives

To determine whether librarian or information specialist authorship is associated with better reproducibility of the search, at least three databases searched, and better reporting quality in dental systematic reviews (SRs).

### Methods

SRs from the top ten dental research journals (as determined by Journal Citation Reports and Scimago) were reviewed for search quality and reproducibility by independent reviewers using two Qualtrics survey instruments. Data was reviewed for all SRs based on reproducibility and librarian participation and further reviewed for search quality of reproducible searches.

### Results

Librarians were co-authors in only 2.5% of the 913 included SRs and librarians were mentioned or acknowledged in only 9% of included SRs. Librarian coauthors were associated with more reproducible searches, higher search quality, and at least three databases searched. Although the results indicate librarians are associated with improved SR quality, due to the small number of SRs that included a librarian, results were not statistically significant.

### Conclusion

Despite guidance from organizations that produce SR guidelines recommending the inclusion of a librarian or information specialist on the review team, and despite evidence showing that librarians improve the reproducibility of searches and the reporting of methodology in SRs, librarians are not being included in SRs in the field of dental medicine. The authors of this review recommend the inclusion of a librarian on SR teams in dental medicine and other fields.

**Competing interests:** The authors have declared that no competing interests exist.

## Introduction

Systematic reviews (SRs) serve an important role in evidence-based practice across the health professions. Initially established within the health disciplines by the founding of the Cochrane Collaboration in the early 1990s, in the ensuing decades SRs have been produced at an exponential rate. The increase in the number of published SRs has not necessarily been marked by an increase in quality. Studies of published SRs indicate major shortcomings, ranging from the failure of reviewers to adequately report methodology [1] to inadequate adherence to standards [2].

In response to these evaluations, international groups have created guidance statements for SRs and meta-analyses. The primary reporting guideline, the Preferred Reporting Items for Systematic reviews and Meta-Analyses (PRISMA) statement, comprises 27 required reporting items and a flow diagram to track the review process [3]. Although widely accepted, meta-research indicates that PRISMA adherence is spotty. Reviews that mention use of PRISMA demonstrate no better adherence to guidelines than reviews without mention of reporting standards [4].

The methodological quality of SRs can be highly variable across health professions [2, 5, 6], despite the widespread use of international guidelines on SRs such as those from the Cochrane Collaboration, the National Academy of Medicine, AHRQ, MOOSE, and others. Many studies of methodological and reporting quality in SRs and meta-analyses come from the domains of medicine, leaving a paucity of knowledge of these issues across other health professions such as dental medicine. More limited studies nonetheless indicate that shortcomings in reporting and methodological quality extend to reviews published in dental medicine [7].

PRISMA use in dentistry SRs varies greatly, promulgated by lack of required reporting standards in journals [8, 9]. Even published dental SRs that assert the use of PRISMA guidelines are often inadequate in adherence, with screening methods underreported and inclusion criteria and data extraction methods missing [10]. Analyses of the methodological quality of SRs in dentistry and endodontics document equally low methodological quality, finding poor reporting and conduct of SRs [10]. Poor adherence to SR guidance is found as early as the registration process for orthodontics SRs [11]. Adherence to other markers of methodological rigor in dental SRs is equally problematic, with issues identified in screening and data extraction processes, inclusion of gray literature, use of quality assessment, and inclusion of a risk of bias assessment [7]. An examination of methodological quality of meta-analyses of periodontal treatment of glycemic control in diabetic patients revealed only 33% of the included meta-analyses met criteria for high methodological quality [12].

The quality of searches underlying dental medicine SRs also suffers from shortcomings. An assessment of the reproducibility of search strategies for 530 dental medicine SRs found that none had complete reporting of the search strategies and selection process [1]. Another study examining SRs in prosthodontic and implant journals found that fewer than 5% of the studies employed 'systematic' searches, with issues identified in the rigor and sensitivity of the searches and lack of inclusion of gray literature [13]. Dental medicine SRs are not unique in search strategy shortcomings. A study of 137 SRs published January, 2018, and identified through the PubMed database indicated a high level of search errors which adversely affected the information retrieved from the searches [14]. A study of Cochrane systematic reviews in 2002 contained similar errors [15].

Various influential organizations supporting SRs and meta-analyses strongly recommend that a librarian or information specialist be included in the research team as a means of addressing such issues. Notably, the Cochrane Handbook for Systematic Reviews of Interventions and the 2011 guidelines on SRs from the Institute of Medicine (now the National

Academy of Medicine) strongly recommend including a librarian or information specialist from the start of the review process [16, 17]. Other guidelines are less specific on timing, but suggest that information specialists be involved [18–21]- recommendations that stem from evidence indicating that the inclusion of librarians in SR teams can help improve the quality of searches, the quality of methodological reporting, and the inclusion of gray literature in the search [22–26]. Additional guidance, directed specifically toward librarians and information specialists, suggests peer review of SR searches and provides the Peer Review of Electronic Search Strategies (PRESS) instrument for evaluation of searches in hopes of improving quality [27]. Although librarian roles in SRs have generally focused on ensuring a rigorous and reproducible search, librarians may be involved in numerous parts of the review [28]. Rethlefsen, et al, take librarian inclusion even further, noting that librarian co-authors were correlated with improved reporting of search strategies in SRs [26]. When examining the International Committee of Medical Journal Editors (ICMJE) recommendations of who should be a named author [29], Rethlefsen's assertion that librarians should be named co-authors makes sense for those SRs that have librarian participation in formulating a search and reporting methodology, among other tasks.

Evidence of the effect of librarian inclusion on the quality of SRs has come from studies utilizing SRs from clinical medicine. Published literature has not documented the extent and effect of the inclusion of librarians on SRs in dental medicine. Given the recognized need for the inclusion of librarians [30], this study aims to address this gap in the literature. The research questions addressed are:

1. Does having a librarian involved improve search strategy reproducibility?

2. When a librarian is involved, does having a librarian as a coauthor improve the reproducibility of the search more than just having the librarian acknowledged?

3. Does librarian involvement in the review improve reporting quality?

4. Does the involvement of a librarian make it more likely that grey literature is searched?

5. Does having a librarian involved lead to the use of at least 3 databases searched?

## Methods

### Database search and selection of SRs

This study utilized published SRs and meta-analyses from the twelve most highly cited dental medicine journals. The top ten journals were selected from two sources: Journal Citation Reports (JCR) and Scimago. As the top ten journals differed between the two journal indices, a compound list of the top ten journals from each database was created and duplicates were removed. The list of journals and the number of articles examined from each journal can be found in S1 Appendix.

One author (KS) conducted a search for SRs in each of these journals within the PubMed database on June 18, 2018. The search string consisted of the PubMed journal title abbreviations combined with the SR search string in the PubMed Clinical Queries tool (prior to the 2020 update to PubMed). The combined search is available in S2 Appendix. No date restrictions were used, but results were limited to English language articles.

The search results were loaded into Rayyan for initial screening. Two authors (JS, KS) independently reviewed the articles to identify SRs or meta-analyses. An article was considered a SR if it met at least one of the following inclusion criteria:

• Article title or abstract indicated the study was a SR or meta-analysis

- The authors described the use of a systematic search

- The authors described the use of multiple databases to identify all relevant literature on a topic

    Exclusion criteria:

- Article explicitly described the methodology as a non-SR expert-level review, such as a scoping review or narrative review, even if a systematic search was employed

- Article described qualitative or quantitative study methods for human subjects (cohort studies, randomized controlled trials, retrospective chart reviews, etc.)

    Conflicts were resolved by discussion. After review, 430 articles were removed, leaving 913 articles included in the analysis.

## Method for reproducibility assessment and data extraction

Articles were randomized using a random sequence generator in Excel and divided into two sets of articles. The two sets were assigned to pairs of reviewers (JS, KS, TE, CF). Assessment was blinded and performed in duplicate using an assessment form in Qualtrics. Any conflicts were resolved by a third reviewer not involved in the initial assessment (JB, JS, KS).

    The assessment instrument was adapted from one used by Rethlefsen and colleagues (2015) in their study of the effect librarian co-authorship had on the quality of reported search strategies in internal medicine SRs. Questions were adapted to yes/no format wherever possible. Questions focused on reporting areas that are recommended in the PRISMA 2009 recommendations. The adapted survey consisted of 18 questions [S3 Appendix]. This assessment survey was used primarily to determine which SRs had reproducible search strategies and which SRs included a librarian or information specialist (subsequently collectively referred to as 'librarians') in any capacity. These primary data points allowed for further analysis of SRs with searches deemed reproducible and SRs with librarian participation. To determine if a paper included a librarian as author or in acknowledgement, reviewers examined names and titles of all included authors and those mentioned in acknowledgements. Where an author had an MLS, MLIS, MSIS or other library degree, or where the text of the methods section or acknowledgement mentioned a librarian, reviewers automatically recorded the librarian's participation as an author or as acknowledged. Where degrees or titles were not provided for authors or those acknowledged, the reviewers made every attempt to determine the author's credentials, from searching the author's name to investigating any affiliated institutions. Other data points of interest in this first analysis included reporting methods listed where the search information was located, if inclusion and exclusion criteria were reported, number of reviewers who conducted each part of the SR screening, the number of articles found in database searching and at each level of screening, and whether a risk of bias assessment was performed. To limit bias, articles were identified using PubMed Identification (PMID) numbers and all articles went through this first assessment before the second assessment was conducted.

    A reproducible search was one that could be copied and pasted into the search bar with minimal changes and produce similar results to those reported in the published SR. Reasonable leeway for reproducibility was considered to be either the absence of a date filter or the inclusion of each concept group in a table, with clear direction on how to combine the concept groups. Where there were questions about the reproducibility of a search through visual assessment, the search was copied and pasted into the database mentioned in text to determine reproducibility. Any articles that used the PubMed database were analyzed using the version of PubMed that was available as default prior to 2020, as the new version of PubMed (2020-present) would

provide different results. Searches were considered irreproducible if they did not: state the database the search was conducted in, specify how parts of the search were combined, or indicate what tags were attached to the search terms (i.e. mentioned the use of MeSH terms and keywords in description, but did not note MeSH terms in the provided search).

After analysis, 452 articles were determined to have reproducible search strategies.

A second evaluation instrument was used to further analyze articles with searches deemed reproducible [S4 Appendix]. The purpose of the second evaluation was to analyze the quality of the reproducible searches. Reviewers were blinded to all previous data points (except search reproducibility) and were unaware of which articles had librarian participation. Data points included the number of databases searched, names of databases searched, if searches were reproducible exactly as written or if they required manipulation, whether search terms were reasonably developed and complete, overall quality of the search (subjective), errors in the use of parentheses and brackets, and overall syntax errors. Reviewers also assessed whether gray literature was searched. For the purpose of this analysis, gray literature included conference abstracts, dissertations and theses, discipline-specific publications that are not indexed in major databases, trials registries, institutional repositories, and other sources outside the scope of major databases. The articles were divided into three sets and three authors (JS, KS, JB) were assigned two of the three sets to review so that each article was independently reviewed by two reviewers. The evaluation was performed using Qualtrics. Conflicts were resolved by a third author. In cases where all three authors disagreed, conflicts were resolved by discussion. To limit bias, articles were again identified using PMID numbers. Reviewers were instructed to only examine the searches in this step and not to check authors or affiliations. Finally, reviewers were blinded to all data points from the first assessment survey.

## Data analysis

Data were combined and coded in Excel and analyzed using SPSS statistical software version 26. Data visualizations were created in Excel and Google Spreadsheets. Data is available via OSF (https://osf.io/qsua8/files/?view_only=ea7a550400f4403eb280b1d383ec11fa). Statistics were collected and analyzed as planned and included descriptive statistics for all data points. Crosstabs with Chi-Square tests for significant differences were used to analyze all data points that included the dependent variable of reproducible searches. Independent variables included whether a librarian assisted or co-authored a review, whether gray literature was searched, number of reviewers who examined titles, abstracts, and full texts, whether reviewers were blinded to each others' work, and whether or not a risk of bias assessment was conducted. Bayesian ANOVA of Likert data (overall quality of the search–subjective score) was also conducted with Chi-Square tests for significance. Any significance was determined by p value < .05 [Table 1].

## Results

### Librarian participation

Few published dental medicine SRs reported the inclusion of a librarian in any capacity on the team. Of the 913 SRs examined, 2.5% (n = 23) included a librarian as a co-author, 9% (n = 82), mentioned or acknowledged a librarian, and in 88.5% of the SRs, inclusion of a librarian in the SR was not reported.

### Inclusion of search strategy

Inclusion of a librarian on the team is associated with improved reporting of at least one search strategy [Fig 1], particularly when a librarian is included as an author. Of the 23 articles with a

**Table 1. Data analysis.**

| Research Question | Associated Questionnaire Question(s) (see S3, S4 Appendices) | Independent Variable(s) (Code) With Response Code(s) | Dependent Variable(s) (Code) With Response Code(s) | Variable Type | Analysis |
|---|---|---|---|---|---|
| Does having a librarian involved improve search strategy reproducibility? | Questionnaire 1 Question 3, 5 Questionnaire 2 Question 3 | Search Reproducibility (SRCHREPROD) 1 = Yes 2 = No | Librarian Participation (LIBPARTICP) 1 = Librarian/Information specialist as author 2 = Librarians/information specialist is mentioned in text or acknowledged 3 = No or unclear | Categorical/ Nominal | Crosstab with Chi Square |
| When a librarian is involved, does having a librarian as a coauthor improve the reproducibility of the search more than just having the librarian acknowledged? | Questionnaire 1 Question 3, 5 Questionnaire 2 Question 3, 5 | Search reproducibility (SRCHREPROD) 1 = Yes 2 = No | Librarian participation (LIBPARTICP) 1 = Librarian/Information specialist as author 2 = Librarians/information specialist is mentioned in text or acknowledged 3 = No or unclear | Categorical/ Nominal | Crosstab with Chi Square |
| Does librarian involvement in the review improve reporting quality? | Questionnaire 1 Question 3–18 Questionnaire 2 Question 3, 4, 6, 7 | Location of search Info. (SRCHLOCINF) 1 = Full search strategy in text 2 = Description in text 3 = In appendix–in article 4 = In appendix–web only 5 = No search information 6 = Contact author for full search strategies 7 = Other **All items below coded as (unless otherwise noted):** 1 = Yes 2 = No Search reproducibility (SRCHREPROD) Inclusion criteria listed (INCCRITINC) Exclusion criteria listed (EXCCRITINC) Were researchers blinded to each others' responses (BLINDINGYN) Was the number of reviewers title/ abstract Reported (REVTANUMRP) Number of reviewers title/abstract (NUMREVWRTA) Was the number of reviewers full text reported (REVFTNUMRP) Number of reviewers full text (NUMREVWRFT) Did they report number of studies included (REVSTNUMRP) Did they report number of duplicates (RPDUPREMVD) Did they report number of titles/ abstracts screened (RPTIABSCRN) Did they report number of full texts screened (RPFLTXSCRN) Did they report number of studies included (RPARTFULIN) Was risk of bias assessed (RSKBIASPRF) 1 = Yes 2 = No or unclear | Librarian participation (LIBPARTICP) 1 = Librarian/Information specialist as author 2 = Librarians/information specialist is mentioned in text or acknowledged 3 = No or unclear | Categorical/ Nominal | Crosstab with Chi Square |
| Does the involvement of a librarian make it more likely that grey literature is searched? | Questionnaire 1 Question 3, 5 Questionnaire 2 Question 6 | Did the reviewers conduct a gray literature search? (DIDGREYLIT) 1 = Yes 2 = No | Librarian Participation (LIBPARTICP) 1 = Librarian/Information specialist as author 2 = Librarians/information specialist is mentioned in text or acknowledged 3 = No or unclear Search Reproducibility (SRCHREPROD) 1 = Yes 2 = No | | |
| Does having a librarian involved lead to the use of at least 3 databases searched? | Questionnaire 1 Question 3, 5 Questionnaire 2 Question 3, 4 | Number of Databases Searched (NUMDBSRCHD) Where is search information located (SRCHLOCINF_ | Librarian Participation (LIBPARTICP) 1 = Librarian/Information specialist as author 2 = Librarians/information specialist is mentioned in text or acknowledged 3 = No or unclear Search Reproducibility (SRCHREPROD) | Categorical/ Nominal | Crosstab with Chi Square |

Indicates how data was analyzed for each question of this review.

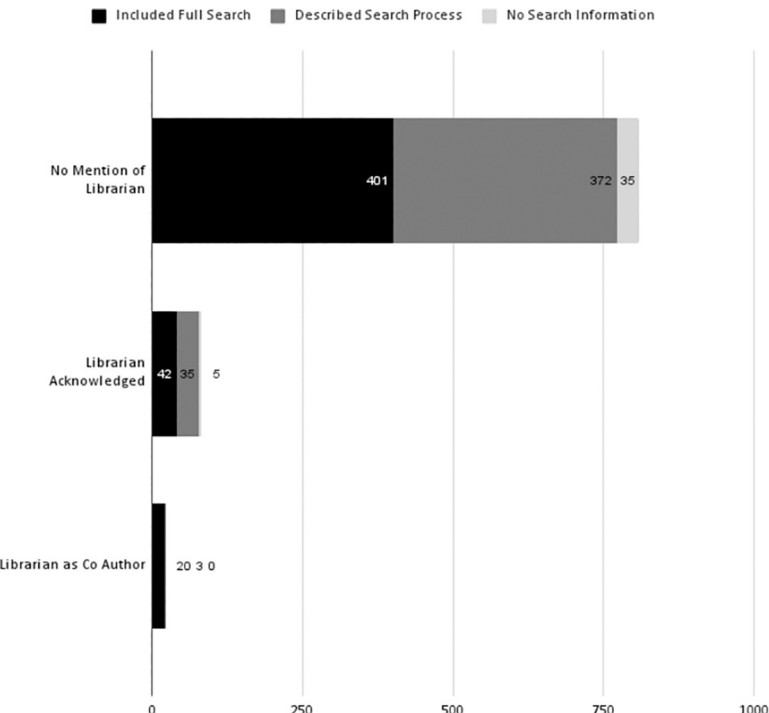

**Fig 1. Inclusion of search strategy.** Illustrates number of articles that included at least one full search, described the search process, or provided no search information.

librarian co-author, 87% (n = 20) included at least one full search strategy in the text or appendix. The remaining 13% (n = 3) provided a description of the search but not a full search strategy. Of the 82 articles that acknowledged or mentioned a librarian, 51% (n = 42) included at least one full search strategy in text or appendix, 43% (n = 35) described the search, and 6% (n = 5) did not include a search strategy or description. Of the 808 articles that did not mention or acknowledge librarian participation, 50% (n = 401) included at least one full search strategy, 46% (n = 372) described the search, and the remaining 4% (n = 35) did not include a search strategy or a description.

## Search quality

Subjective analysis of overall quality of the search indicated that librarian participation on the review team increased the quality of the search. Each reviewer provided a score (1–5). When scores were within 2 points, average score was calculated. When there was a spread of more than 2 points, a third reviewer scored the article and all 3 scores were averaged. Based on Bayesian ANOVA assessment of Likert quality scoring 1(low) - 5(high), when a librarian was a coauthor, the mean score was 3.5. When a librarian was acknowledged, the mean score was 3.4. When a librarian was not mentioned, the mean score was 2.814.

## Reporting in all reviews

**Information reported in reviews.** Among the 913 reviews, only 13.3% (n = 121) of the reviewers reported and searched for grey literature. We conducted initial analysis of five additional reporting metrics: reporting of inclusion and exclusion criteria, blinding during the review, reporting the number of title/abstract reviewers, and reporting the number of full text

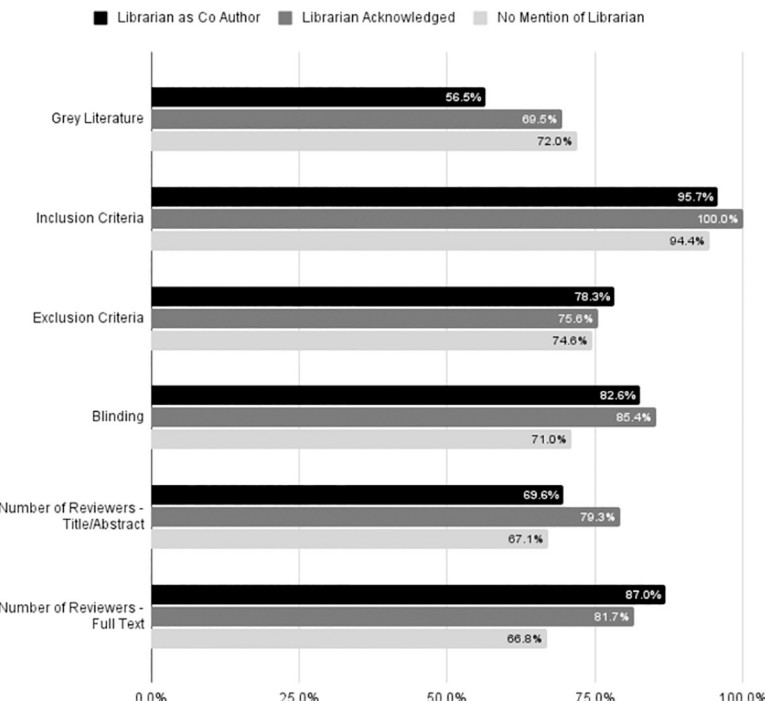

**Fig 2. Information reported in reviews.** Illustrates the percent of all articles examined that included the information listed. Grey Literature was reported in 56.5% of reviews with a librarian co-author, 69.5% of reviews with an acknowledged librarian, and 72% of articles with no mention of a librarian. Inclusion criteria was reported in 95.7% of reviews with a librarian co-author, 100% of reviews with an acknowledged librarian, and 94.4% of articles with no mention of a librarian. Exclusion criteria was reported in 78.3% of reviews with a librarian co-author, 75.6% of reviews with an acknowledged librarian, and 74.6% of articles with no mention of a librarian. Whether authors were blinded to each other's work was reported in 82.6% of reviews with a librarian co-author, 85.4% of reviews with an acknowledged librarian, and 71% of articles with no mention of a librarian. Number of reviewers who examined titles and abstracts of articles was reported in 69.6% of reviews with a librarian co-author, 79.3% of reviews with an acknowledged librarian, and 67.1% of articles with no mention of a librarian. Number of reviewers who examined full texts of articles was reported in 87% of reviews with a librarian co-author, 81.7% of reviews with an acknowledged librarian, and 66.8% of articles with no mention of a librarian.

reviewers. On average, only 1.2 of these items were included. Of the 913 articles reviewed, only 33 (3.6%) included all five reporting factors. Almost half (46%) did not report any of these metrics [Fig 2].

**Databases.** Librarian participation was associated with at least 3 databases searched, as the Cochrane Handbook for Systematic Reviews of Interventions recommends the use of the following databases in Systematic Reviews: CENTRAL, MEDLINE, and Embase (where available as this is a subscription database) [17]. Of the 913 SRs, only 42% (n = 386) specified the number of databases searched. Reviews that included a librarian co-author included a mean average of 5.2 databases searched and 88% (n = 14/16) reported searching at least three sources. Reviews that acknowledged or mentioned a librarian included an average of 4.7 databases and 91% (n = 31/34) reported searching at least three sources. Reviews that were unclear about librarian participation included an average of 3.6 databases searched and 71% (n = 240/336) reported searching in at least three sources.

When examining only the reproducible searches that reported the number of databases searched (n = 385), the numbers are similar. When a librarian was a co-author, 88% (n = 14/16) reported searching at least three sources. When a librarian was acknowledged, 91%

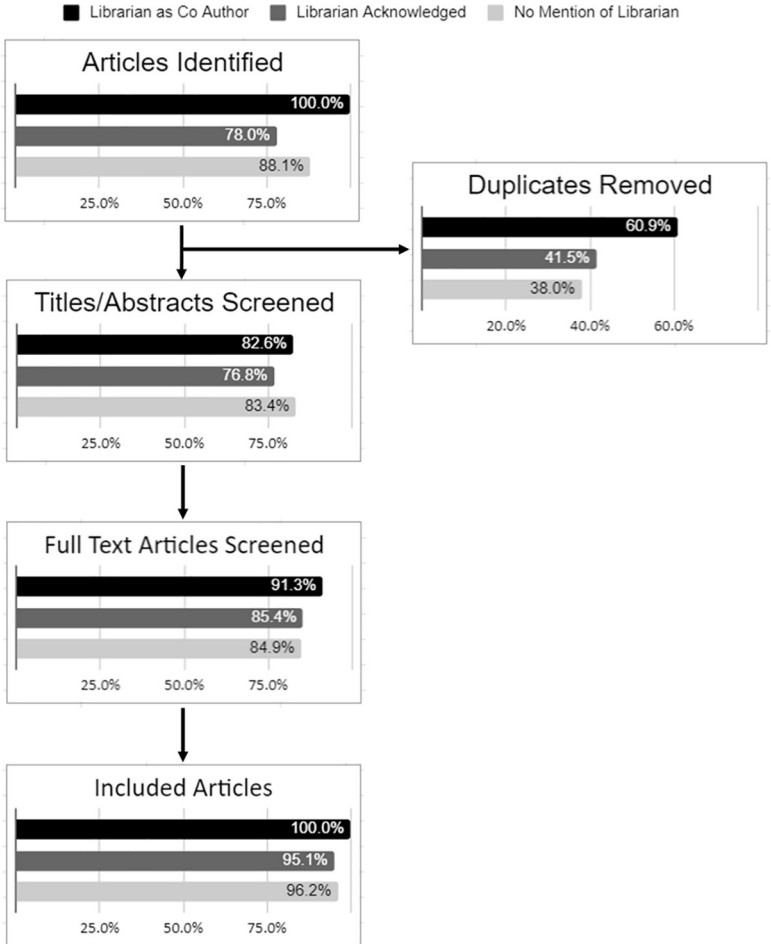

**Fig 3. Numbers as reported in reviews.** Illustrates the percent of articles that reported the following: number of articles identified from database searching, number of duplicates removed, number of titles and abstracts screened, number of full text articles screened, and the number of articles included in the final review. Each of these was broken down by whether a librarian was included as a co-author, if a librarian was acknowledged, or if there was no mention of a librarian.

(n = 31/34) reported searching at least three sources. When there was no report of a librarian involved, 71% (n = 239/335) reported searching at least three sources.

**Reporting the numbers.** We examined five metrics that are commonly reported within the methodology of a SR. A librarian co-author was associated with better reporting of numbers of articles identified through searching, duplicates removed, full texts screened, and articles included in the analysis. In only one category—number of titles/abstracts screened—did groups without librarian participation include better reporting [Fig 3]. On average, articles reported 1.1 of these metrics and only 31 (3.4%) of the 913 articles reported all five metrics. Over one third of the reviews (38%) reported none of these metrics.

**Reproducibility.** Librarian co-authors are associated with a higher rate of reproducible searches. There were 23 articles that listed a librarian as an author. Of those, 69.6% (n = 16) were reproducible. When a librarian or information specialist was acknowledged, 51.2% (n = 42) of the 82 articles were reproducible. Among the articles that did not acknowledge a librarian as an author or contributor, 48.8% (394) were reproducible.

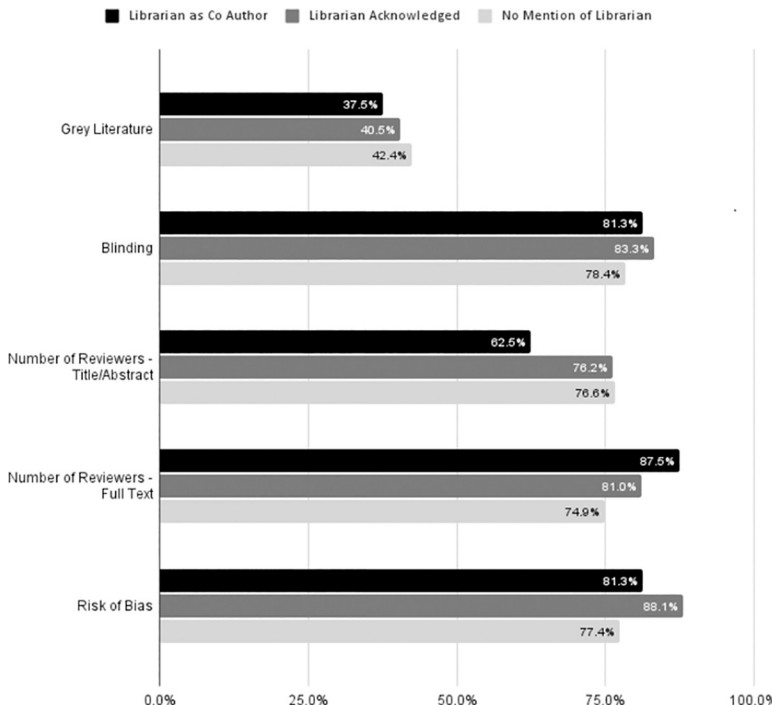

**Fig 4. Information reported in reproducible searches.** Within reproducible searches, illustrates the percent of articles that included the information listed. Grey Literature was reported in 37.5% of reviews with a librarian co-author, 40.5% of reviews with an acknowledged librarian, and 42.4% of articles with no mention of a librarian. Whether authors were blinded to each other's work was reported in 81.3% of reviews with a librarian co-author, 83.3% of reviews with an acknowledged librarian, and 78.4% of articles with no mention of a librarian. Number of reviewers who examined titles and abstracts of articles was reported in 62.5% of reviews with a librarian co-author, 76.2% of reviews with an acknowledged librarian, and 76.6% of articles with no mention of a librarian. Number of reviewers who examined full texts of articles was reported in 87.5% of reviews with a librarian co-author, 81% of reviews with an acknowledged librarian, and 74.9% of articles with no mention of a librarian. Whether a risk of bias assessment was completed was reported in 81.3% of reviews with a librarian co-author, 88.1% of reviews with an acknowledged librarian, and 77.4% of articles with no mention of a librarian.

## Analysis of reproducible searches

The subset of articles with at least one reproducible search was further analyzed using SPSS statistical software version 26. Where multiple searches were reported, the first search strategy (usually MEDLINE via PubMed or Ovid) was analyzed. In articles where the first search strategy was not reproducible, the first reproducible search was analyzed.

**Librarian assistance.** Crosstabulation analysis revealed, of the 452 articles with reproducible searches, only 3.5% (n = 16) specified a librarian or information specialist as an author. In 9.3% (n = 42) a librarian was acknowledged. The other 87.2% did not specify the inclusion of a librarian.

## Information reported in reproducible searches

Librarian co-authors were not associated with more frequent reporting of whether grey literature was searched or how many reviewers examined titles and abstracts of articles. Librarians were, however, associated with higher rates of blinding among reviewers, reporting the number of reviewers who examined full texts of articles, and risk of bias assessments on included articles [Fig 4].

**Search details.** Librarian co-authors were associated with fewer mistakes in the use of brackets, Boolean operators, and proximity operators. Spelling was also slightly better with a

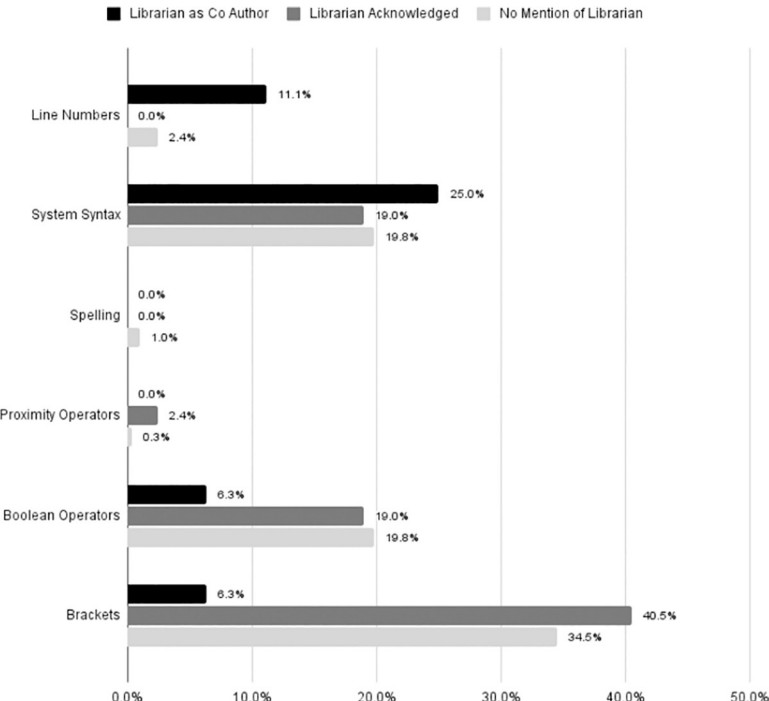

**Fig 5. Search syntax mistakes.** Within reproducible searches, illustrates the percent of articles with errors in the following categories: Line numbers (only searches with line numbers were assessed), System syntax errors, Spelling errors, Errors in the use of proximity operators (only searches with proximity operators were assessed), Errors in the use of Boolean operators, Errors in the use of brackets.

librarian co-author. The inclusion of a librarian co-author was not associated with fewer errors in system syntax or line numbering [Fig 5]. Speculation about the reason for higher librarian mistakes in this section will be addressed in the discussion.

When the search terms were examined, librarian co-authors were associated with fewer heading mistakes and irrelevant headings, fewer missing spelling variants, and about equal use of unwarranted limits when compared to other reviews; however, librarian co-authors were associated with more missing and irrelevant natural language terms and less optimal use of truncation [Fig 6].

**Significance.** Analysis of most of our data points in SRs did not demonstrate statistical significance (p < .05). This is likely due to the small sample size of SRs that included librarians in any capacity and the even smaller proportion of reproducible searches that included a librarian. We were, however, able to demonstrate statistical significance when examining the effect of a librarian on whether at least three databases were searched. Based on chi square testing, when a librarian was a co-author, it was 11 times more likely that at least 3 databases were searched than when a librarian was not a co-author with a p value of .003.

## Discussion

### Principal findings

There is very little evaluation of SRs in the field of dental medicine, however our findings are consistent with existing research from other disciplines. Despite guidelines from the Cochrane Collaboration, AHRQ, and CRD suggesting consulting a librarian on a systematic review search is valuable [3, 13, 16–19], inclusion of librarians in dental medicine SRs is extremely

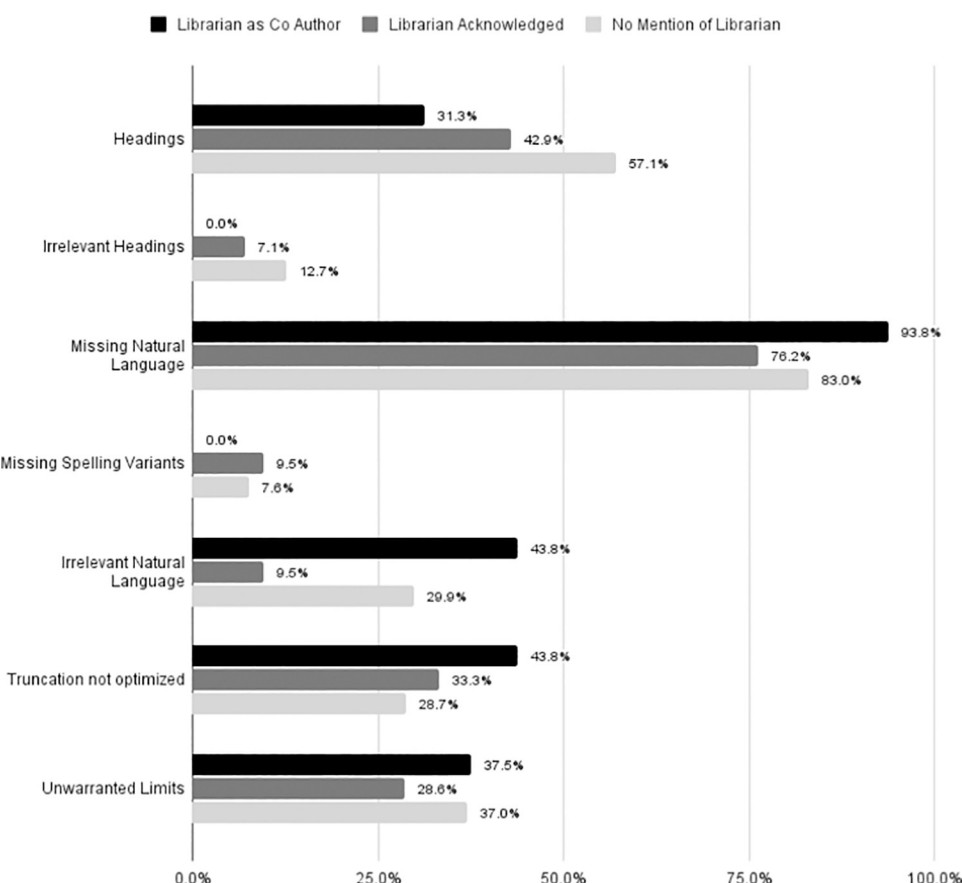

**Fig 6. Search language mistakes.** Within reproducible searches, illustrates the percent of articles with errors in the following categories: Use of MeSH or Emtree headings (only searches with headings were assessed), Inclusion of irrelevant MeSH or Emtree headings (only searches with headings were assessed), Missing natural language variants, Missing spelling variants (such as pediatric/paediatric), Inclusion of irrelevant natural language variants, Errors in the use of truncation (only searches with truncation were assessed), Inclusion of unwarranted limits (date, language, etc.).

limited. Reporting of methodology in dental medicine is not ideal [7–10]. Our findings suggest that having a librarian as part of the review team is associated with improved methodological reporting of SRs, consistent with similar foundational studies in other disciplines, such as those by Koffel and Rethlefsen [23–27]. Although subjective, analysis of search quality indicates librarian co-authorship is also associated with a better-quality search, supporting prior research in medicine [23–26]. Librarian co-authors were associated with a higher inclusion rate of grey literature, although looking only at SRs with reproducible searches, librarian co-authors were not associated with a higher rates of grey literature inclusion. We notice numerous librarian errors in the search details section (Fig 5). We posit two possible reasons for errors in line numbers, system syntax, and proximity operators. Although statistical analysis did not focus on these data points, we suggest that SR teams with librarian inclusion were more likely to utilize a search strategy that included line numbers and proximity operators than SR teams without librarian assistance. This possibility should be examined further, as this could have skewed the data. Another possible reason for errors in this section is that librarians are more likely, because of their familiarity with a search or database, to neglect or overlook reporting certain data. More research is needed in this area as well, although librarians should be more vigilant about checking their wording in the methods section. Due to the small sample

size and not knowing the extent of librarian roles, it is unclear whether librarian co-authorship is associated with improved reporting of numbers; however, when taken together, articles with librarian co-authorship and librarian participation performed overall better than those without a librarian.

## Implications

A sensitive search in the SR process takes on special importance because it determines the set of articles on which all other parts of the review depend. As articles were collected prior to the publication of PRISMA-S guidelines in 2021 [31], those conducting SRs had only their knowledge of PRISMA 2009 guidelines and search methodology to guide them. Given the methodological knowledge and search expertise a librarian could add to a SR team, this raises questions of why they are not being utilized on dental SR teams. One issue could be the level of librarian support available at dental schools, private practices, or hospitals. For instance, the Medical Library Association (MLA) dental caucus only has 93 members while other caucuses have up to 500 members. There are 68 accredited dental schools in the United States, and 10 in Canada [32]. Even a designated librarian for a dental school may not be able to support all systematic reviews due to workloads. While there may be enough support for the 78 dental schools in the US and Canada, librarian assistance outside of academia may be problematic. Even within academia, not all librarians have formal SR training.

The explosion of publications of SRs of low quality is well documented [6]. As many authors have concluded, there is a lack of awareness in the broader field of dentistry on the methodology and execution of an SR and what is required for a truly sensitive search. When journals do not require use of reporting standards, the number of high quality reviews in dentistry research decreases [7]. The lack of reproducible searches found could indicate a lack of knowledge of what is needed to re-run a search, limiting the ability to validate and update the research.

## Limitations

There were some limitations to this review. First, the rate of librarian assistance may be higher than reported in articles, and therefore, data may be skewed [23]. Second, most authors that acknowledged a librarian did not explain the librarian's role. Due to the yes/no formatting of many questions, there is no allowance for degrees or levels of compliance; therefore, for example, an SR that was missing only a couple of keywords had the same result as one that was missing many. The subjective rating of the search quality was an attempt to adjust for this weakness; however, this is an area that would benefit from further study and analysis.

Although many data points were collected about the quality of reporting and search terms, overall quality of searches was determined through subjective rating. Reviewers limited bias where possible by being blinded to each other's work. Articles were identified using PMID numbers within search instruments, and the two-step review process with librarian participation evaluated in the first step and search quality evaluated in the second allowed for blinding to which articles were deemed to have librarian participation when reviewers assessed search quality. The number of SRs was substantial; however, the number with reproducible searches and librarian participation was too small to run meaningful statistical significance analysis. Searches in SRs were evaluated prior to the release of PRISMA-S guidelines for searches [31] and PRISMA 2020 reporting guidelines [33]. The publication of these new guidelines may positively affect the quality of future searches and the quality of reporting in SRs, and should be investigated.

## Conclusion

Overall findings suggest that involvement of a librarian is associated with more reproducible searches and improved methodological reporting in dental medicine SRs, though the association in the current review is not statistically significant. Efforts should be made to consult a librarian and include them as a co-author or acknowledge them in the article for their assistance in crafting sensitive, reproducible searches and for their knowledge of SR methodology and reporting standards.

## Supporting information

**S1 Appendix.**
(DOCX)

**S2 Appendix.**
(DOCX)

**S3 Appendix.**
(DOCX)

**S4 Appendix.**
(DOCX)

## Acknowledgments

The authors thank Dr. Hui Bian for her recommendations and expertise on assessment tool creation and statistical analysis. The authors also thank Dr. Julia Castleberry for her recommendations and expertise on statistical analysis.

## Author Contributions

**Conceptualization:** Jana Schellinger, Kerry Sewell.

**Data curation:** Jana Schellinger, Kerry Sewell, Jamie E. Bloss.

**Formal analysis:** Jana Schellinger.

**Investigation:** Jana Schellinger, Kerry Sewell, Jamie E. Bloss, Tristan Ebron, Carrie Forbes.

**Methodology:** Jana Schellinger, Kerry Sewell.

**Validation:** Jana Schellinger.

**Writing – original draft:** Jana Schellinger, Kerry Sewell, Jamie E. Bloss.

**Writing – review & editing:** Jana Schellinger, Kerry Sewell, Jamie E. Bloss, Tristan Ebron, Carrie Forbes.

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
