## [Decision Letter · Decision Letter 0]

22 Jun 2021

PONE-D-21-15877

The effect of librarian involvement on the quality of systematic reviews in dental medicine

PLOS ONE

Dear Dr. Bloss,

Thank you for submitting your manuscript to PLOS ONE. After careful consideration, we feel that it has merit but does not fully meet PLOS ONE’s publication criteria as it currently stands. Therefore, we invite you to submit a revised version of the manuscript that addresses the points raised during the review process.

We look forward to receiving your revised manuscript.

Kind regards,

Lisa Susan Wieland

Academic Editor

PLOS ONE

Journal Requirements:

Could you therefore please include the title page into the beginning of your manuscript file itself, listing all authors and affiliations

Reviewers' comments:

Reviewer's Responses to Questions

**Comments to the Author**

1. Is the manuscript technically sound, and do the data support the conclusions?

Reviewer #1: Yes

Reviewer #2: Partly

2. Has the statistical analysis been performed appropriately and rigorously? 

Reviewer #1: Yes

Reviewer #2: I Don't Know

3. Have the authors made all data underlying the findings in their manuscript fully available?

Reviewer #1: Yes

Reviewer #2: Yes

4. Is the manuscript presented in an intelligible fashion and written in standard English?

Reviewer #1: Yes

Reviewer #2: Yes

5. Review Comments to the Author

Reviewer #1: This is an interesting and well-researched paper, illustrating the often poor reporting of literature search methods in dental systematic reviews. The writing is clear and the data is well-presented, with limitations acknowledged and neatly described.

I have some minor comments for the authors:

1. Lines 43-44: the statement that: "More limited studies nonetheless indicate that shortcomings in reporting and methodological quality extend to reviews published in dental medicine" - I would expect to see some referencing to support this (I don't doubt that is true!). For example, El-Rabbany M, Li S, Bui S, Muir JM, Bhandari M, Azarpazhooh A. A Quality Analysis of Systematic Reviews in Dentistry, Part 1: Meta-Analyses of Randomized Controlled Trials. J Evid Based Dent Pract. 2017 Dec;17(4):389-398. doi: 10.1016/j.jebdp.2017.06.004. Epub 2017 Jun 28. PMID: 29197440. This paper reports some improvement in dental SRs, but also that "there is considerable room for improvement" still. It would support this statement I think.

2. How did the team determine that the SR authors were/were not librarians or information scientists? Was this simply down to self-reporting of the job titles in the author line? I'm guessing so, but it would be good to have this spelled out explicitly.

3. Is there any data to support the assertion in lines 84-85 that there is a correlation between the number of databases searched and the quality of the search? It may be reasonable to search only selected databases depending on the topic and there may be reduced efficiency / some search redundancy involved in searching a large number of databases if this is not warranted. It may be interesting to look at a subset of reviews which searched more than four databases and check that wider searching was justified by the topic. I believe that information scientists are generally moving away from the assumption that more databases searched is equal to a higher quality search, see the paper by Bramer et al, which instead considers optimal databases to search: Bramer, W.M., Rethlefsen, M.L., Kleijnen, J. et al. Optimal database combinations for literature searches in systematic reviews: a prospective exploratory study. Syst Rev 6, 245 (2017). https://doi.org/10.1186/s13643-017-0644-y.

4. Lines 151-154: This paragraph is a little confusing. What do you mean by "significance" in this context? What exactly was being measured?

Reviewer #2: Thank you for the opportunity to review this paper. This topic is of enormous interest to me, and I am happy to see that further research is being conducted in this area. In this article, the authors examine systematic reviews published in dental journals to assess the role and impact of librarians, the reproducibility of searches, and the quality of searches. This work builds upon prior work by others in clinical medical fields. The sheer volume of assessed manuscripts and searches in this work is impressive; having conducted similar research myself, I know that assessing search quality, for example, can take multiple hours for one search.

After reading this article, I have several questions and comments that I hope can help the authors strengthen this impactful manuscript.

1. Based on the content of the manuscript, I was surprised not to see several key papers cited and referenced, including the two major papers on errors in search strategies (Salvador-Olivan et al 2019 and Sampson & McGowan 2006) and PRESS (McGowan et al 2016). It seems that these might bring possibilities for direct comparisons, if not just references.

2. Regarding the text, Finding What Works in Health Care, the authors (correctly) note that it was published by the Institute of Medicine. Since the name of that organization has changed, it may be beneficial to readers to note the new name as well.

3. Line 75: The authors note that previous studies focused on internal medicine, yet they also cite relevant work looking at pediatrics, surgery, and cardiology. Perhaps this should be “clinical medicine” instead of “internal medicine”?

4. The authors do not discuss the role of librarian authorship in the introduction specifically, yet it is one of the primary research questions. Perhaps it would be helpful to add a brief sentence describing why this would be important in the intro?

5. Lines 95-99: The authors do not say when the search took place, and neither do they comment anywhere in the article on the time period included in their search and/or results. It would be helpful to state the dates of the searches as well as that there was no limit in date (if that is the case).

6. Line 97: The authors clearly conducted their search prior to the new revision of the PubMed systematic reviews subset search. It might be helpful to note that this was the active query at the date of the search; it is completely different today.

7. Lines 129-36: It was not clear to me; did the authors actually copy and paste in the searches to test reproducibility? It appears they may have, but this is not spelled out. If they did, this would be the first study to do so, which should be noted.

8. Lines 137-143: It was helpful that the study instrument was linked here, but I admit that I did not review the instrument immediately; I was therefore greatly surprised by many of the results, which did not appear to be mentioned in the manuscript. I would suggest that the main data elements, especially those reported in the results, should be listed and/or briefly described in the manuscript.

9. Lines 137-43: When assessing searches for quality, were the authors blinded to: the authors, the presence of a librarian, the findings from the first review, etc? The practice of duplicate assessment would reduce some of the inherent bias, certainly, but when librarians are assessing other librarians’ work, there may unconscious bias at play (both in that we would expect more from each other, so we might be pickier, and the opposite, that we would rate quality higher because it would match our expectations).

10. Lines 146-48: How did the authors conduct their analysis? What statistics were planned to be used? What were the planned analyses (i.e., multivariate regression? Linear regression?)? I see one location (line 173) where the authors mention one statistical analysis used, but this is the only place. For example, see the primary comparative articles that are similar topics for their analytical methods: Koffel et al, Meert et al, Rethlefsen et al.

11. It would be helpful to have other information about the data set in the results; information about years included, number of reviews per journal title, etc, could potentially help readers get a better grasp on what is actually being studied.

12. Lines 151-154: I appreciate that the authors are being up front that there is no statistical significance here, but I think that this should be located further down in the results—and the type of analysis used should be described in the methods.

13. Line 163 (and Lines 129-43): The authors do not make it explicitly clear whether they assessed whether ALL searches are available, or whether having AT LEAST ONE search available counts for the measure, “included the full search strategy.” This is a very different metric, and while I suspect it was “at least one” and not “all” (based on line 168, e.g., saying “a full search strategy”), this would be helpful to comment upon, as would the change in requirements for reporting between PRISMA 2009 and PRISMA 2020/PRISMA-S.

14. Lines 172-6: There is nothing in the appendices or the methods that I can see that discusses using an overall scored quality assessment. Am I missing it? This would definitely need to be discussed, along with details about how reviewer concordance/interrater reliability was assessed.

15. Lines 224-33: Because the authors’ data collection instrument uses binary responses to challenging questions, I think it is really difficult to say whether degree of librarian involvement led to anything. For example, one article without a librarian mentioned might use a single keyword for a search, missing potentially multiple synonyms. An article with a librarian author might have 10 keyword synonyms for a concept, but could miss one. They would have the same results in this study, however, because there is not a measurement of degree. I think it would be informative for the authors to discuss this as a limitation. I also think it would be really difficult to assess these for over 400 articles, in duplicate—pretty impressive! For example, Salvador-Olivan et al only examined 137 articles. It’s clear the scale of this work necessitated this simpler method.

16. Lines 238-46: I’m not entirely convinced that the authors’ conclusion matches their findings; to me, I can’t see any effect of librarian co-authorship on search quality (though clearly there is an association in the inclusion of a full search). Are the authors weighing some of their findings higher than others? It seems to be about evenly split between those areas where librarian co-authors are better and worse.

17. Discussion section: There is no comparison with other research anywhere in the Discussion section, which makes it hard to understand the context of these findings. Do they differ from other disciplines? Do they match other (non-librarians’) studies done in dental medicine? I would really like to see an expanded discussion section here, as I think this is an important study that would benefit from contextualization.

18. Line 269-70: The authors could cite Koffel here (#21) as evidence that others have shown that librarians are not always mentioned, even when they played a significant role.

19. Line 272: The authors suggest that bias was “limited where possible,” but I’m not clear on how that happened—was it just reviewing in duplicate?

20. Lines 277-8. I would suggest restating this sentence as, “Overall findings suggest that involvement of a librarian is associated with more reproducible searches and improved methodological reporting in dental medicine systematic reviews, though the association is non-significant.”

21. The Appendices lack titles, which makes it difficult to quickly identify.

22. Appendix D: What is meant by a grey lit search? How did the authors define it (this should really be in the Methods section)?

23. Figures: I may be missing something, but do the figures have captions? It was really difficult to figure out what I was looking at just by looking at the figure.

24. Fig 1: I’m not sold on how this figure was composed. I find it really confusing, personally. I think it would be more effective to use all of the search data here (including no search), as that would more starkly show the differences—and wouldn’t necessitate readers figuring out that the 100% only refers to 2 of the 3 collected data points.

25. Figures: Fig 2 and 3 use black for “no mention of a librarian.” The remaining figures use black for “librarian as co-author.” This is incredibly confusing.

26. Fig. 3: Why did the authors analyze librarian participation prior to deduplication? Seems odd.

27. Fig 5: Are the numbers here referring to number of articles with errors in line numbers? Or it looks like percentages of articles? Did most of the articles even use line numbers? How did you account for not applicable responses? From this graph, it appears pretty clearly that librarians make lots and lots of errors, except with brackets and Boolean, compared to non-specialists.

6. PLOS authors have the option to publish the peer review history of their article (what does this mean?). If published, this will include your full peer review and any attached files.

Reviewer #1: **Yes: **Anne Littlewood

Reviewer #2: No

---

## [Author Response · Author response to Decision Letter 0]

25 Jun 2021

AUTHOR RESPONSE: Thank you sincerely to the two reviewers of our manuscript entitled The effect of librarian involvement on the quality of systematic reviews in dental medicine. We appreciate your valuable feedback and the extensive amount of time and thought you put into it. We have attempted to address your questions and concerns as follows (all page numbers refer to the Revised Manuscript with Track Changes):

Reviewer #1: This is an interesting and well-researched paper, illustrating the often poor reporting of literature search methods in dental systematic reviews. The writing is clear and the data is well-presented, with limitations acknowledged and neatly described.

I have some minor comments for the authors:

1. Lines 43-44: the statement that: "More limited studies nonetheless indicate that shortcomings in reporting and methodological quality extend to reviews published in dental medicine" - I would expect to see some referencing to support this (I don't doubt that is true!). For example, El-Rabbany M, Li S, Bui S, Muir JM, Bhandari M, Azarpazhooh A. A Quality Analysis of Systematic Reviews in Dentistry, Part 1: Meta-Analyses of Randomized Controlled Trials. J Evid Based Dent Pract. 2017 Dec;17(4):389-398. doi: 10.1016/j.jebdp.2017.06.004. Epub 2017 Jun 28. PMID: 29197440. This paper reports some improvement in dental SRs, but also that "there is considerable room for improvement" still. It would support this statement I think.

AUTHOR RESPONSE: Citation added - #11

2. How did the team determine that the SR authors were/were not librarians or information scientists? Was this simply down to self-reporting of the job titles in the author line? I'm guessing so, but it would be good to have this spelled out explicitly.

AUTHOR RESPONSE: Added details in lines 146-153

3. Is there any data to support the assertion in lines 84-85 that there is a correlation between the number of databases searched and the quality of the search? It may be reasonable to search only selected databases depending on the topic and there may be reduced efficiency / some search redundancy involved in searching a large number of databases if this is not warranted. It may be interesting to look at a subset of reviews which searched more than four databases and check that wider searching was justified by the topic. I believe that information scientists are generally moving away from the assumption that more databases searched is equal to a higher quality search, see the paper by Bramer et al, which instead considers optimal databases to search: Bramer, W.M., Rethlefsen, M.L., Kleijnen, J. et al. Optimal database combinations for literature searches in systematic reviews: a prospective exploratory study. Syst Rev 6, 245 (2017). https://doi.org/10.1186/s13643-017-0644-y.

AUTHOR RESPONSE: Added reasoning to lines 99-101. Also added new Figure 3 based off new data.

4. Lines 151-154: This paragraph is a little confusing. What do you mean by "significance" in this context? What exactly was being measured?

AUTHOR RESPONSE: Relocated this section and clarified in lines 357-363

Reviewer #2: Thank you for the opportunity to review this paper. This topic is of enormous interest to me, and I am happy to see that further research is being conducted in this area. In this article, the authors examine systematic reviews published in dental journals to assess the role and impact of librarians, the reproducibility of searches, and the quality of searches. This work builds upon prior work by others in clinical medical fields. The sheer volume of assessed manuscripts and searches in this work is impressive; having conducted similar research myself, I know that assessing search quality, for example, can take multiple hours for one search.

After reading this article, I have several questions and comments that I hope can help the authors strengthen this impactful manuscript.

1. Based on the content of the manuscript, I was surprised not to see several key papers cited and referenced, including the two major papers on errors in search strategies (Salvador-Olivan et al 2019 and Sampson & McGowan 2006) and PRESS (McGowan et al 2016). It seems that these might bring possibilities for direct comparisons, if not just references.

AUTHOR RESPONSE: Added citations and added comparisons in discussion section 367-392

2. Regarding the text, Finding What Works in Health Care, the authors (correctly) note that it was published by the Institute of Medicine. Since the name of that organization has changed, it may be beneficial to readers to note the new name as well.

AUTHOR RESPONSE: Noted new name in lines 71-72

3. Line 75: The authors note that previous studies focused on internal medicine, yet they also cite relevant work looking at pediatrics, surgery, and cardiology. Perhaps this should be “clinical medicine” instead of “internal medicine”?

AUTHOR RESPONSE: Changed wording in line 83

4. The authors do not discuss the role of librarian authorship in the introduction specifically, yet it is one of the primary research questions. Perhaps it would be helpful to add a brief sentence describing why this would be important in the intro?

AUTHOR RESPONSE: Included in lines 68-86

5. Lines 95-99: The authors do not say when the search took place, and neither do they comment anywhere in the article on the time period included in their search and/or results. It would be helpful to state the dates of the searches as well as that there was no limit in date (if that is the case).

AUTHOR RESPONSE: Inserted search date in line 112. Statement about no date restrictions is in line 114.

6. Line 97: The authors clearly conducted their search prior to the new revision of the PubMed systematic reviews subset search. It might be helpful to note that this was the active query at the date of the search; it is completely different today.

AUTHOR RESPONSE: Inserted search date in line 112. Added wording about old version of PubMed in lines 166-168.

7. Lines 129-36: It was not clear to me; did the authors actually copy and paste in the searches to test reproducibility? It appears they may have, but this is not spelled out. If they did, this would be the first study to do so, which should be noted.

AUTHOR RESPONSE: Added wording in lines 164-165.

8. Lines 137-143: It was helpful that the study instrument was linked here, but I admit that I did not review the instrument immediately; I was therefore greatly surprised by many of the results, which did not appear to be mentioned in the manuscript. I would suggest that the main data elements, especially those reported in the results, should be listed and/or briefly described in the manuscript.

AUTHOR RESPONSE: Added wording in lines 153-157 and 176-180.

9. Lines 137-43: When assessing searches for quality, were the authors blinded to: the authors, the presence of a librarian, the findings from the first review, etc? The practice of duplicate assessment would reduce some of the inherent bias, certainly, but when librarians are assessing other librarians’ work, there may unconscious bias at play (both in that we would expect more from each other, so we might be pickier, and the opposite, that we would rate quality higher because it would match our expectations).

AUTHOR RESPONSE: Added wording in lines 158-159, 175-176, 184-186, 423-428.

10. Lines 146-48: How did the authors conduct their analysis? What statistics were planned to be used? What were the planned analyses (i.e., multivariate regression? Linear regression?)? I see one location (line 173) where the authors mention one statistical analysis used, but this is the only place. For example, see the primary comparative articles that are similar topics for their analytical methods: Koffel et al, Meert et al, Rethlefsen et al.

AUTHOR RESPONSE: Adjusted language in lines 191-197.

11. It would be helpful to have other information about the data set in the results; information about years included, number of reviews per journal title, etc, could potentially help readers get a better grasp on what is actually being studied.

AUTHOR RESPONSE: Unfortunately, we do not have data about number of reviews per journal. Added language to data analysis section for more clarity.

12. Lines 151-154: I appreciate that the authors are being up front that there is no statistical significance here, but I think that this should be located further down in the results—and the type of analysis used should be described in the methods.

AUTHOR RESPONSE: Relocated this section and clarified in lines 354-361. Data analysis further described in lines 191-197.

13. Line 163 (and Lines 129-43): The authors do not make it explicitly clear whether they assessed whether ALL searches are available, or whether having AT LEAST ONE search available counts for the measure, “included the full search strategy.” This is a very different metric, and while I suspect it was “at least one” and not “all” (based on line 168, e.g., saying “a full search strategy”), this would be helpful to comment upon, as would the change in requirements for reporting between PRISMA 2009 and PRISMA 2020/PRISMA-S.

AUTHOR RESPONSE: Adjusted language in lines 210-219 and line 302

14. Lines 172-6: There is nothing in the appendices or the methods that I can see that discusses using an overall scored quality assessment. Am I missing it? This would definitely need to be discussed, along with details about how reviewer concordance/interrater reliability was assessed.

AUTHOR RESPONSE: Adjusted language in lines 139-141 to assist in clarity about the modification of the Rethlefsen instrument and language to lines 225-227 for clarity.

15. Lines 224-33: Because the authors’ data collection instrument uses binary responses to challenging questions, I think it is really difficult to say whether degree of librarian involvement led to anything. For example, one article without a librarian mentioned might use a single keyword for a search, missing potentially multiple synonyms. An article with a librarian author might have 10 keyword synonyms for a concept, but could miss one. They would have the same results in this study, however, because there is not a measurement of degree. I think it would be informative for the authors to discuss this as a limitation. I also think it would be really difficult to assess these for over 400 articles, in duplicate—pretty impressive! For example, Salvador-Olivan et al only examined 137 articles. It’s clear the scale of this work necessitated this simpler method.

AUTHOR RESPONSE: Thank you for your recognition of the scope of this work. Language was added to lines 422-428 in limitations to address lack of scale.

16. Lines 238-46: I’m not entirely convinced that the authors’ conclusion matches their findings; to me, I can’t see any effect of librarian co-authorship on search quality (though clearly there is an association in the inclusion of a full search). Are the authors weighing some of their findings higher than others? It seems to be about evenly split between those areas where librarian co-authors are better and worse.

AUTHOR RESPONSE: Adjusted language in the Principal findings section in lines 367-392.

17. Discussion section: There is no comparison with other research anywhere in the Discussion section, which makes it hard to understand the context of these findings. Do they differ from other disciplines? Do they match other (non-librarians’) studies done in dental medicine? I would really like to see an expanded discussion section here, as I think this is an important study that would benefit from contextualization.

AUTHOR RESPONSE: Added language to lines 367-392.

18. Line 269-70: The authors could cite Koffel here (#21) as evidence that others have shown that librarians are not always mentioned, even when they played a significant role.

AUTHOR RESPONSE: Added citation for Koffel article in line 418.

19. Line 272: The authors suggest that bias was “limited where possible,” but I’m not clear on how that happened—was it just reviewing in duplicate?

AUTHOR RESPONSE: Added wording about bias for clarity in lines 158-159, 175-176, 184-186, 423-428.

20. Lines 277-8. I would suggest restating this sentence as, “Overall findings suggest that involvement of a librarian is associated with more reproducible searches and improved methodological reporting in dental medicine systematic reviews, though the association is non-significant.”

AUTHOR RESPONSE: Adjusted language to lines 438-440.

21. The Appendices lack titles, which makes it difficult to quickly identify.

AUTHOR RESPONSE: Added appendix titles

22. Appendix D: What is meant by a grey lit search? How did the authors define it (this should really be in the Methods section)?

AUTHOR RESPONSE: Added information about grey literature in lines 93-96.

23. Figures: I may be missing something, but do the figures have captions? It was really difficult to figure out what I was looking at just by looking at the figure.

AUTHOR RESPONSE: Added legends for figures. Titles and legends are contained in article instead of in figure files as per PlosOne Instructions: https://journals.plos.org/plosone/s/figures#loc-captions

24. Fig 1: I’m not sold on how this figure was composed. I find it really confusing, personally. I think it would be more effective to use all of the search data here (including no search), as that would more starkly show the differences—and wouldn’t necessitate readers figuring out that the 100% only refers to 2 of the 3 collected data points.

AUTHOR RESPONSE: Adjusted Figure 1. See figure file.

25. Figures: Fig 2 and 3 use black for “no mention of a librarian.” The remaining figures use black for “librarian as co-author.” This is incredibly confusing.

AUTHOR RESPONSE: Adjusted figures for consistency.

26. Fig. 3: Why did the authors analyze librarian participation prior to deduplication? Seems odd.

AUTHOR RESPONSE: Figure 3 has become Figure 4. It is purely about reporting of numbers and not about analysis. The use of the PRISMA Diagram format was due to familiarity of the document. Added wording to the figure legend for clarity.

27. Fig 5: Are the numbers here referring to number of articles with errors in line numbers? Or it looks like percentages of articles? Did most of the articles even use line numbers? How did you account for not applicable responses? From this graph, it appears pretty clearly that librarians make lots and lots of errors, except with brackets and Boolean, compared to non-specialists.

AUTHOR RESPONSE: Figure 5 has become figure 6. Speculation about reasons for errors was added in lines 380-389. 

6. PLOS authors have the option to publish the peer review history of their article (what does this mean?). If published, this will include your full peer review and any attached files.

AUTHOR RESPONSE: No thank you.

Do you want your identity to be public for this peer review? For information about this choice, including consent withdrawal, please see our Privacy Policy.

Reviewer #1: Yes: Anne Littlewood

Reviewer #2: No

---

## [Decision Letter · Decision Letter 1]

22 Jul 2021

PONE-D-21-15877R1

The effect of librarian involvement on the quality of systematic reviews in dental medicine

PLOS ONE

Dear Dr. Bloss,

Thank you for submitting your manuscript to PLOS ONE. After careful consideration, we feel that it has merit but does not fully meet PLOS ONE’s publication criteria as it currently stands. Therefore, we invite you to submit a revised version of the manuscript that addresses the points raised during the review process.

Please address both the reviewer clarifications of previous comments and the new comments regarding reorganizing text and providing citations.

We look forward to receiving your revised manuscript.

Kind regards,

Lisa Susan Wieland

Academic Editor

PLOS ONE

Journal Requirements:

Reviewers' comments:

Reviewer's Responses to Questions

**Comments to the Author**

1. If the authors have adequately addressed your comments raised in a previous round of review and you feel that this manuscript is now acceptable for publication, you may indicate that here to bypass the “Comments to the Author” section, enter your conflict of interest statement in the “Confidential to Editor” section, and submit your "Accept" recommendation.

Reviewer #2: (No Response)

2. Is the manuscript technically sound, and do the data support the conclusions?

Reviewer #2: Yes

3. Has the statistical analysis been performed appropriately and rigorously? 

Reviewer #2: I Don't Know

4. Have the authors made all data underlying the findings in their manuscript fully available?

Reviewer #2: Yes

5. Is the manuscript presented in an intelligible fashion and written in standard English?

Reviewer #2: Yes

6. Review Comments to the Author

Reviewer #2: I thank the authors for their excellent work addressing reviewer comments, and I appreciate the chance to review their revised version. This version addresses nearly all of the comments, and the manuscript now offers more context to readers and offers clarity on a number of areas. I have a few remaining questions and comments.

1. The authors do not discuss the role of librarian authorship in the introduction specifically, yet it is one of the primary research questions. Perhaps it would be helpful to add a brief sentence describing why this would be important in the intro?

The authors have added additional content to the introduction, but I am not seeing any addition of information specifically about librarian co-authorship. I still think it would be helpful to mention something specifically about authorship prior to the research questions. The same is true with grey literature, which is a research question, but is not specifically addressed in the intro.

2. Line 97: The authors clearly conducted their search prior to the new revision of the PubMed systematic reviews subset search. It might be helpful to note that this was the active query at the date of the search; it is completely different today.

This item was unfortunately written poorly by me. I meant that the systematic review subset search has changed, not PubMed (though that is also true). If the authors compare the current PubMed systematic reviews search with the old one, they will see it has significantly changed. However, with the date included, it will help readers understand that this was the subset search at the time of that search. Unfortunately, NLM doesn’t seem to track versions of this search anywhere that I’ve seen.

3. Lines 146-48: How did the authors conduct their analysis? What statistics were planned to be used? What were the planned analyses (i.e., multivariate regression? Linear regression?)? I see one location (line 173) where the authors mention one statistical analysis used, but this is the only place. For example, see the primary comparative articles that are similar topics for their analytical methods: Koffel et al, Meert et al, Rethlefsen et al.

I appreciate the additional detail provided by the authors about their analyses. I think there is still more work to be done here. “Mostly” should be avoided. What variables were tested using chi squares, e.g.

4. It would be helpful to have other information about the data set in the results; information about years included, number of reviews per journal title, etc, could potentially help readers get a better grasp on what is actually being studied. AUTHOR RESPONSE: Unfortunately, we do not have data about number of reviews per journal. Added language to data analysis section for more clarity.

I think the authors do have this information, or I would hope so? Do you not have the citations for the included studies?

5. Discussion section: There is no comparison with other research anywhere in the Discussion section, which makes it hard to understand the context of these findings. Do they differ from other disciplines? Do they match other (non-librarians’) studies done in dental medicine? I would really like to see an expanded discussion section here, as I think this is an important study that would benefit from contextualization.

A few new comments:

6. The authors very nicely lay out new research questions. The associated detail with those questions, however, should be in the methods section. Lines 93-6; 99-101

7. PRISMA 2009, PRISMA 2020, and PRISMA-S are mentioned in several places in the text, but the latter two are not cited. It would be appropriate to add citations. (Lines 397; 432-3)

8. I could be missing something, but it looks to me like the information in lines 272-5 is redundant with the information in the paragraph above—with one major caveat. The caveat is that, in line 270, I think 71% is supposed to be 74%.

9. Lines 307-310: The authors may want to consider moving this up to be their first data point in the results, as the analyses do all seem to depend upon it.

7. PLOS authors have the option to publish the peer review history of their article (what does this mean?). If published, this will include your full peer review and any attached files.

Reviewer #2: No

---

## [Author Response · Author response to Decision Letter 1]

5 Aug 2021

Plos ONE Rebuttal Document – Version 2

We again appreciate the thoughtful comments made by Reviewer 2. We appreciate the time and effort taken to help us improve this paper for publication. We believe these adjustments have significantly improved the paper. Thank you.

1. The authors do not discuss the role of librarian authorship in the introduction specifically, yet it is one of the primary research questions. Perhaps it would be helpful to add a brief sentence describing why this would be important in the intro?

The authors have added additional content to the introduction, but I am not seeing any addition of information specifically about librarian co-authorship. I still think it would be helpful to mention something specifically about authorship prior to the research questions. The same is true with grey literature, which is a research question, but is not specifically addressed in the intro.

Added wording about Gray Literature to lines 76-77 and wording about co-authors to lines 82-87 (including a reference [29] to the ICMJE recommendations for authorship).

2. Line 97: The authors clearly conducted their search prior to the new revision of the PubMed systematic reviews subset search. It might be helpful to note that this was the active query at the date of the search; it is completely different today.

This item was unfortunately written poorly by me. I meant that the systematic review subset search has changed, not PubMed (though that is also true). If the authors compare the current PubMed systematic reviews search with the old one, they will see it has significantly changed. However, with the date included, it will help readers understand that this was the subset search at the time of that search. Unfortunately, NLM doesn’t seem to track versions of this search anywhere that I’ve seen.

Added wording to lines 113-114 about the use of the Clinical Queries search string prior to the PubMed update. Also, complete search is included in Appendix A, with the Clinical Queries search contained in the full search.

3. Lines 146-48: How did the authors conduct their analysis? What statistics were planned to be used? What were the planned analyses (i.e., multivariate regression? Linear regression?)? I see one location (line 173) where the authors mention one statistical analysis used, but this is the only place. For example, see the primary comparative articles that are similar topics for their analytical methods: Koffel et al, Meert et al, Rethlefsen et al.

I appreciate the additional detail provided by the authors about their analyses. I think there is still more work to be done here. “Mostly” should be avoided. What variables were tested using chi squares, e.g.

Added wording to Data Analysis section (lines 193-204) and Table 1

4. It would be helpful to have other information about the data set in the results; information about years included, number of reviews per journal title, etc, could potentially help readers get a better grasp on what is actually being studied. AUTHOR RESPONSE: Unfortunately, we do not have data about number of reviews per journal. Added language to data analysis section for more clarity.

I think the authors do have this information, or I would hope so? Do you not have the citations for the included studies?

As all studies were identified with PMID numbers, this was not a data point we originally ran. Re-ran the journal title search compared to all included PMID numbers in response to this comment. Added number of examined articles from each journal to Appendix A.

5. Discussion section: There is no comparison with other research anywhere in the Discussion section, which makes it hard to understand the context of these findings. Do they differ from other disciplines? Do they match other (non-librarians’) studies done in dental medicine? I would really like to see an expanded discussion section here, as I think this is an important study that would benefit from contextualization.

A few new comments:

6. The authors very nicely lay out new research questions. The associated detail with those questions, however, should be in the methods section. Lines 93-6; 99-101

Moved associated detail to the methods section.

7. PRISMA 2009, PRISMA 2020, and PRISMA-S are mentioned in several places in the text, but the latter two are not cited. It would be appropriate to add citations. (Lines 397; 432-3)

Added references for PRISMA 2020 and PRISMA-S

8. I could be missing something, but it looks to me like the information in lines 272-5 is redundant with the information in the paragraph above—with one major caveat. The caveat is that, in line 270, I think 71% is supposed to be 74%.

Adjusted language for clarity. Now lines 273-284

9. Lines 307-310: The authors may want to consider moving this up to be their first data point in the results, as the analyses do all seem to depend upon it.

We have arranged the paper with two main Results sections: Reporting in All Reviews; Analysis of Reproducible Searches. We have arranged the data point as the first in the section labeled: Analysis of Reproducible Searches.

---

## [Decision Letter · Decision Letter 2]

17 Aug 2021

The effect of librarian involvement on the quality of systematic reviews in dental medicine

PONE-D-21-15877R2

Dear Dr. Bloss,

We’re pleased to inform you that your manuscript has been judged scientifically suitable for publication and will be formally accepted for publication once it meets all outstanding technical requirements.

Kind regards,

Lisa Susan Wieland

Academic Editor

PLOS ONE

Additional Editor Comments (optional):

Reviewers' comments:

Reviewer's Responses to Questions

**Comments to the Author**

1. If the authors have adequately addressed your comments raised in a previous round of review and you feel that this manuscript is now acceptable for publication, you may indicate that here to bypass the “Comments to the Author” section, enter your conflict of interest statement in the “Confidential to Editor” section, and submit your "Accept" recommendation.

Reviewer #2: All comments have been addressed

2. Is the manuscript technically sound, and do the data support the conclusions?

Reviewer #2: Yes

3. Has the statistical analysis been performed appropriately and rigorously? 

Reviewer #2: I Don't Know

4. Have the authors made all data underlying the findings in their manuscript fully available?

Reviewer #2: Yes

5. Is the manuscript presented in an intelligible fashion and written in standard English?

Reviewer #2: Yes

6. Review Comments to the Author

Reviewer #2: (No Response)

7. PLOS authors have the option to publish the peer review history of their article (what does this mean?). If published, this will include your full peer review and any attached files.

Reviewer #2: No

---

## [Editor Report · Acceptance letter]

23 Aug 2021

PONE-D-21-15877R2 

The effect of librarian involvement on the quality of systematic reviews in dental medicine 

Dear Dr. Bloss:

I'm pleased to inform you that your manuscript has been deemed suitable for publication in PLOS ONE. Congratulations! Your manuscript is now with our production department. 

Kind regards, 

on behalf of

Dr. Lisa Susan Wieland 

Academic Editor

PLOS ONE